# Gender, Age, Family and Territorial Features of Dietary and Physical Activity Patterns in Russian Youths

**DOI:** 10.3390/ijerph19095779

**Published:** 2022-05-09

**Authors:** Marina B. Kotova, Sergey A. Maksimov, Oksana M. Drapkina

**Affiliations:** National Medical Research Center for Preventive Medicine of the Ministry of Healthcare of the Russian Federation, 10 Bld. Petroverigskiy Lane, 101990 Moscow, Russia; mkotova@gnicpm.ru (M.B.K.); drapkina@bk.ru (O.M.D.)

**Keywords:** adolescents, diet, daily routine, physical activity, family social status, family smoking, regional conditions

## Abstract

Objective: Identifying dietary and physical activity (PA) patterns in Russian youths and examining their dependence on gender, age, family characteristics and area of residence features. The research involved 783 school students 10–17 years of age and their parents living in the cities of Moscow and Murmansk. Using the principal component analysis, four integral indices were identified that characterized the habits and dietary patterns, PA and lifestyle. Boys, compared with girls, were more likely to consume unhealthy foods but less likely to practice malnutrition. Within the age ranges of 13–15 years old and 16–17 years old, the proportions of school students with a poor sleep pattern and low PA were higher than in children 10–12 years of age. In smoking families, children were less likely to consume healthy foods and more likely to eat unhealthy foods. In Murmansk school students, compared with their Moscow peers, a reduction in both sleep time and PA was observed less often. Our study demonstrated that the most significant factors of a balanced and healthy diet, rational daily routine and lifestyle in school students were their gender and age characteristics, as well as some contextual factors.

## 1. Introduction

Modern lifestyles and global economic and digital transformations in developed countries increase the risks associated with chronic non-communicable diseases. A significant contribution to the global burden of disease, disability and mortality is made by unhealthy eating habits and PA [1]. Lack of PA, malnutrition and sedentary lifestyle are closely interrelated and, often formed in childhood and adolescence, persist into adulthood [2,3], having a significant impact on health at a later age [1,2,4,5]. For instance, malnutrition in childhood increases the risk of developing obesity and non-communicable diseases in adult life, such as hypertension, cardiovascular disease, diabetes, certain types of cancer and various atopic diseases [4,6,7,8,9]. Family, social environment and school are the most important sources of education about a healthy lifestyle. Living conditions and upbringing in the family are among the most important factors in the formation of children’s health [7,10,11]. In addition, the food preferences and eating habits of parents, combined with the culture of eating, are modeled and assimilated, first of all, at home [7,10,11,12,13,14].

The high pace of change in the contemporary world, associated over the past decades with globalization, urbanization and migration, has had a profound impact on family traditions, lifestyles and eating habits and has led many children to spend more time outside their homes from an early age [7,15,16,17,18,19,20]. Population-based studies, PeNSE [21] and ERICA [22], demonstrated negative trends in the diet and eating regimen of adolescents, such as reduced consumption of fruits and vegetables and increased consumption of sugary drinks and ultra-pasteurized foods. The habit of skipping meals, especially breakfast, eating late at night, erratic eating, eating large amounts of fast food and processed foods, and an increase in snacking are most common in adolescence [4,23,24,25,26,27]. According to various studies, up to 10–30% of children and youths never eat breakfast and up to 75% of them regularly skip the morning meal [28,29,30]. At the same time, children who eat breakfast regularly have a higher quality of nourishment with the right amount of consumption of key food groups, such as fruits, vegetables, dairy products and dietary fiber. Children who skip breakfast, on the contrary, tend to eat more high-calorie foods, such as fast food, which leads to overeating [4,30,31,32,33,34].

The lack of PA in school students is indicated by many recent studies worldwide. Progressively more children spend their free time watching TV, playing computer games or surfing the Internet, which negatively affects the health of the younger generation, manifesting itself in various physical problems or psychological complaints [35,36]. In many countries, the proportion of children with insufficient PA exceeds 75%, especially among girls and older youths [36,37,38].

In addition to individual and family factors, diet and PA may depend on the regional characteristics of the population and related cultural, traditional and climatic characteristics [39,40,41]. Many studies suggested that not only in adults, but also in children, individual and family characteristics did not fully explain the territorial differences in the prevalence of behavioral risk factors [42,43,44,45]. The Russian Federation is a large state in terms of territory, with significantly different geographical, social and economic living conditions of the population, along with ethnic and cultural differences that may influence the formation of lifestyle and behavioral factors, which was previously shown for the adult population [20,46]. No such studies involved Russian children.

The objective of our study was identifying dietary and physical activity (PA) patterns in Russian youths and examining their dependence on gender, age, family characteristics and area of residence features. At the first stage, the patterns of eating habits and PA of adolescents were formed. At the second stage, associations of the obtained patterns with individual gender and age characteristics, social conditions of living in the family, and the place (city) of residence were studied. The main hypothesis of our study was the assumption that individual (gender- and age-related) characteristics, and family and contextual (city) living environment affected the dietary patterns and patterns of PA in Russian youths.

## 2. Materials and Methods

### 2.1. Study Design and Sampling

We conducted the cross-sectional study from September through November of 2014 among school students aged 10–17 years in four public secondary schools in Moscow and two schools in Murmansk. The choice of schools represented convenience sampling; the choice of students in schools represented continuous sampling. Schools were selected based on the possibility of conducting a study given an approval by the governments of Moscow and Murmansk, as well as by the school administrations. To ensure comparability of data from Moscow and Murmansk, the choice of schools was carried out in residential areas that were similar in terms of the socioeconomic living conditions of the population. Schools were located in areas that were similar in terms of the socioeconomic living conditions of the population. All teenagers in grades V-VI and VIII-X in the selected schools and their parents were invited to participate in our study. Written informed consent to participate in the examination and interview of students, a questionnaire survey of their parents, and the use of essential medical documentation was obtained in advance. The total response was 78%, virtually the same by city.

Two school students were excluded from the general sample due to the lack of reliable data about their ages. For some indicators (social living conditions, dietary pattern, physical activity), there were also some missing data, less than 5%. Based upon initial values of gender, age and city of residence, the missing data were imputed using the k-nearest neighbors algorithm. Hence, the final sample size with complete data (baseline and reconstructed) was 783 adolescents. The general characteristics of the sample are presented in Table 1.

The study was performed in compliance with Good Clinical Practice and the principles of the Declaration of Helsinki. The study protocols were approved by the Ethics Committee at the National Research Center for Therapy and Preventive Medicine of the Russian Ministry of Healthcare (Moscow, Russia).

### 2.2. Data Collection Tools

Information about the family social status was obtained by questioning parents: family composition (incomplete, i.e., only one parent; complete), number of children in the family (one child; two or more children), mother’s education (not higher; higher), family smoking (none; yes), living conditions of a child participating in the study (separate room; room shared with someone else from the family members).

Dietary and PA patterns were specified via interviewing children. The development of questions and answer options was based on previous domestic and foreign studies on the lifestyles of school-age youths [47,48,49]. Food consumption was assessed using a frequency questionnaire, with the conversion of consumption frequencies into scores (number of points). The frequencies of consuming fruits and vegetables, fast food, hot meals (excluding fast food), carbonated drinks, meat products (including poultry), fish and fish products, milk and dairy products were grouped into four grades: daily consumption (once a day or more times per day) = 1 point; 3–4 times a week = 2 points; once a week or less = 3 points; none at all = 4 points. The consumption of smoked meats and canned foods was assessed via two grades: none or not sure = 1 point; yes = 2 points. The frequency of meals was assessed via two grades: 1–2 times a day or varying day by day = 2 points; 3–5 times a day = 1 point. The presence of a morning meal was assessed sensu five grades: daily = 1 point; only on school days = 2 points; only on weekends = 3 points; rarely = 4 points; none = 5 points. Hence, an increase in points indicates unfavorable trends in dietary patterns: a decline in the frequency of consumption of healthy foods (save for fast food and carbonated drinks), a rise in the frequency of canned food consumption, along with a reduction in the frequency of meals and availability of breakfasts.

Sleep time was quantified in hours and minutes. The question about morning exercises presumed three answer options: none = 3 points; once in a while = 2 points; always = 1 point. Attending school physical education lessons was evaluated according to two grades: no = 2 points; yes = 1 point. The question on the incidence of outdoor walks presumed three answer options: daily = 1 point; only on weekends = 2 points; rarely (staying mostly at home) = 3 points. Thus, an increase in sleep duration was interpreted as a favorable trend, whereas an increase in response scores for the other three questions was considered a negative trend.

A detailed description of all independent and dependent variables is provided in Appendix A.

### 2.3. Assessment of Diet and Physical Activity

Diet and PA characteristics are strongly interrelated, both contextually and statistically. To reduce the dimensionality of the data, the principal component analysis was employed, conventionally used to identify sum indices based on several baseline characteristics [20,50]. To simplify the structure of factors and improve their interpretability, varimax orthogonal rotation was used. Factors were identified when the share of explained variance exceeded 5%, followed by a gap assessment sensu the Cattell scree plot. Overall, four integral factors were identified (Table 2). We would like to specify those as follows. Adherence to DP1 (dietary pattern) characterizes a reduction in the consumption of fruits and vegetables, meat, fish, dairy products and hot meals; accordingly, we call this pattern *rare consumption of a balanced diet*. Adherence to DP2 specifies an increase in the consumption of fast food, soda and canned food; that is why we call it *frequent consumption of unhealthy foods*. Adherence to DP3 characterizes malnutrition, i.e., insufficient food intake per day or an absence or rarity of breakfasts. Adherence to PAP (PA pattern) can be interpreted as a decrease in sleep time and PA in children. For each participant in the study, a quantitative assessment of his predisposition to the identified patterns was obtained with a mean value in the sample of 0.00 and a standard deviation of 1.00. The range of values for DP1 is from −2.15 to 5.53, for DP2 from −2.41 to 3.37, for DP3 from −1.43 to 3.07, and for PAP from −2.87 to 2.78. The higher the magnitude of the child’s adherence to the pattern, the worse his/her habits are in terms of dietary and PA patterns. All identified patterns were grouped into tertiles depending on the individual commitment of the children.

To assess the stability of the identified DPs and PAP, the principal component analysis was carried out in the total sample, as well as separately by gender and age groups, and also by cities (see Appendix A). When comparing factor solutions, Tucker’s congruence coefficient was employed [51]. In general, the factor solutions were similar; therefore, by analogy with other studies [20,50], we decided to use finite factor loadings using the varimax rotation and the selection of four factors from the total sample.

### 2.4. Statistical Data Processing

We present categorical data as percentages. Two-way associations of categorical variables were assessed using Pearson’s Chi-squared test. The effect of gender, age, place of residence and family living conditions on the likelihood of children falling into unfavorable patterns in terms of their dietary and PA patterns was assessed via logistic regression. In the course of the statistical analysis, the third tertiles of DP1, DP2, DP3, and PAP were considered unfavorable. Doing so, an adjustment was made for all individual indicators used in the study: gender, age, city, availability of a separate room, family size, number of children in the family, mother’s education and family smoking. Our results are expressed via odds ratios (OR) and 95% confidence intervals (CI). In order to better understand the associations obtained in the general sample, similar analyses were carried out in strata by gender, age (10–12 years old, 13–15 years old, 16–17 years old) and city of residence. Principal component analysis was performed using the Statistica, version 10.0 (Statsoft Inc., Tulsa, OK, USA). Pearson’s Chi-squared test and logistic regression analysis were performed in SPSS, version 22 (IBM Corp., Armonk, NY, USA). The critical level of statistical significance was assumed at *p* = 0.05.

## 3. Results

### 3.1. Two-Way Associations of the Proportion of Children in Adverse Patterns

Two-way associations of children falling into the third tertiles of dietary patterns and PA pattern depending on their gender, age, city of residence and social conditions are presented in Table 3. Among boys, compared with girls, the proportion of those who often consumed unhealthy foods was higher, but the proportion of those who adhered to malnutrition was lower, 44.2% and 23.2%, respectively (*p* < 0.0001); and 29.5% and 38.9%, correspondingly (*p* = 0.0055). Children 13–15 years of age, compared with 10–12 y/o, more often pursued malnutrition, 39.1% and 27.9%, respectively (*p* = 0.021). At 13–15 years old and at 16–17 years old, the proportion of youths with a poor sleep pattern and low PA increased, compared with 10–12 y/o: 40.6%, 47.3% and 11.6%, respectively (*p* < 0.0001). Children living in incomplete families more often adhered to an unbalanced diet, compared with their peers living with both parents, 42.0% and 32.0%, respectively (*p* = 0.018). In families with one of the family members smoking, compared with smokefree families, there were higher shares of children who rarely consumed healthy foods (37.8% and 28.9%, *p* = 0.0099) and often consumed unhealthy foods (40.9% and 29.8%, *p* = 0.0015).

### 3.2. Effect of Gender, Age and Place of Residence on the Likelihood of Adverse Patterns

In the general sample (Table 4), boys had a higher probability of DP2 than girls (2.72; 1.97–3.75) and a smaller odds ratio of DP3 (0.65; 0.48–0.88). At the same time, the association of gender in children with DP2 differed depending on their age (Appendix A). If at 13–15 y/o and at 16–17 y/o, the associations were similar to the general sample, then in the 10–12 y/o category, on the contrary, the probability of DP2 in boys was lower than in girls (0.48; 0.23–0.99). In addition, when living in Murmansk, boys had a lower probability of DP1 (0.55; 0.34–0.90).

We detected age associations in the general sample by DP3 and PAP (Table 4). Compared with 10–12 y/o youths, 13–15 y/o adolescents were more likely to have DP3 (1.63; 1.12–2.38) and PAP (5.69; CI 3.55–9.13). Children aged 16–17 vs. 10–12 y/o were more likely to have PAP (7.34; 4.45–12.10). When stratified ( Appendix A), it became clear that associations of age with DP3 were typical solely for girls, but not for boys, and also only for children living in Moscow, but not in Murmansk. In addition, 16–17 y/o girls had a higher probability of DP1 (2.14; 1.16–3.94).

In the general sample, place of residence was associated with PAP (Table 4): Murmansk youths exhibited an inverse relationship (0.71; 0.51–0.99), compared with Moscow teenagers. Stratification analysis ( Appendix A) demonstrated that such an association was typical just for girls, but not for boys, and only for children aged 13–15, but not for other age groups. Moreover, only at the age of 10–12 years, Murmansk children had a higher probability of DP2 (3.64; 1.84–7.20), compared with their Moscow peers.

### 3.3. Effect of Social Conditions on the Likelihood of Adverse Patterns

The presence/absence of a separate room for a child in the family did not exhibit associations with dietary and PA patterns in the general sample (Table 4). When stratified ( Appendix A), only girls without a separate room displayed a high probability of DP2 (1.75; 1.01–3.05). In addition, solely children from Murmansk without a separate room had a higher probability of DP3 (2.04; 1.14–3.65).

In the general sample, children living in a complete family had a lower probability of DP3 (0.66; 0.45–0.96). This association was statistically significant only for children aged 10–12, but not for older children (Supplementary Table 3). Moreover, only Moscow children showed an inverse association with DP2 (0.49; 0.29–0.82), and only Murmansk children demonstrated an inverse association with PAP (0.50; 0.28–0.91).

No associations were detected between dietary and PA patterns with the number of children in the family. However, stratification ( Appendix A) revealed inverse associations of the number of children in the family with DP2 only in girls (0.54; 0.32–0.92), and only in 16–17 y/o children (0.49; 0.26–0.92).

A mother’s higher education was inversely associated with the probability of DP2 in children in the general sample (0.69; 0.48–0.98), compared with non-tertiary education (Table 4). This association was statistically significant only in girls 10–12 and 16–17 years old ( Appendix A). Solely girls exhibited an inverse association of a mother’s education with DP3 (0.57; 0.35–0.94). Moreover, the reverse association of a mother’s education with DP3 was typical for children living in Murmansk (0.61; 0.37–0.99).

In the general sample, family smoking was directly associated with DP1 (1.47; 1.07–2.03) and DP2 (1.64; 1.18–2.27), compared with smokefree families (Table 4). An association with DP1 was more typical for boys, but not for girls, and an association with DP2 occurred in children 10–12 years old and 13–15 years old, but not in 16–17 years old ( Appendix A). When stratified by place of residence, it became obvious that both associations, DP1 and DP2, were typical for Moscow adolescents, but not for Murmansk children. Stratification also revealed individual associations that were not typical for the general sample. Thus, the probability of DP3 in smoking families was higher in children aged 12–15 (1.74; 1.07–2.81), as well as in children living in Murmansk (2.07; 1.26–2.42). In addition, an inverse association with PAP was observed in children 16–17 years of age (0.55; 0.31–0.97) and in children living in Moscow (0.50; 0.32–0.79).

## 4. Discussion

This study analyzed data from a survey of school students in two Russian cities, Moscow and Murmansk. The main results of the study were the associations of four integral dietary and PA patterns in school students (DP1, DP2, DP3 and PAP) with gender, age, socioeconomic and demographic characteristics (mother’s education, family smoking, living conditions of school students), as well as the region of residence (Arctic region: Murmansk; West-Central region: Moscow). Of the studied patterns, DP2 and DP3 were consistently associated with the gender of adolescents, while PAP was associated with their age.

### 4.1. Gender-Specific and Age-Specific Dietary Features

At the age of 13–17 years, boys are more committed to the *frequent consumption of unhealthy foods* (DP2), which corresponds to the gender and age specificities of adolescence, when, against the background of changes in hormonal levels, they experience more intense physical development. Male adolescents at this age are more active, which affects the amount of consumed food and the choice of food, including fast food and sugary carbonated drinks, characterized by high calorie content, excessive content of saturated fats, trans-fatty acids, sugar and salt [4,8,31,52].

Unlike boys, the association with DP2 in girls manifests itself at a younger age (10–12 years), since, on the one hand, growing up and, accordingly, going beyond the traditional family dietary framework occurs in girls earlier than in boys. Despite the fact that unhealthy foods are included in the diet of girls at an earlier age (10–12 years), already from the age of 13, they start focusing on healthy, and therefore better-quality, foods due to a more conscious approach to their appearance and health [53,54,55]. In addition, in a cultural context, slimness is more important for girls than for boys [56,57]. Grooming in adolescent females is an element of beauty and femininity and a strong motivation that affects their dietary preferences between the ages of 14 and 17 years old. This is enhanced and reinforced by the influence of society and an immediate environment, which has the greatest impact precisely in adolescence [58,59].

It should be noted that dependence on social group norms, including the negative influence of peers associated with the frequent consumption of high-calorie, low-nutrient foods, and the easy availability of unhealthy foods near schools are key factors that adolescents from different cultures believe affect their eating behavior [60,61,62]. For some adolescents, emotional factors, including stress and depressed mood, can also influence eating behavior via increasing the consumption of fast foods and sugary carbonated drinks (DP2) and poorer diet (DP3) [63,64]. Even though we did not study the influence of emotional factors on dietary patterns, it is possible that rapid hormonal changes, new relationships, contacts, a heightened perception of stress and other characteristics of this age period may be the key factors influencing the eating behavior of adolescents, thereby increasing their consumption of snacks, sweets and the total amount of food [61,62,64,65]. Overall, our data matched the results of studies conducted in other countries, showing that with age, boys were more likely to increase their consumption of unhealthy foods, while girls, due to gender characteristics and stable social behaviors, were less likely to consume fast food, high-sugar food, and other high-calorie unhealthy food products, apparently focusing on maintaining beauty and attractiveness [66,67].

Our data also implied that girls were more prone to malnutrition (DP3): to a smaller incidence of daily breakfasts or their absence, and to a reduced frequency of meals. Indeed, despite the fact that eating breakfast is considered an important component of a healthy lifestyle, according to studies conducted worldwide, with the onset of adolescence, there is an increasing tendency to skip breakfast, especially among girls [29,30,34,52,68]. It is the regularity of eating breakfast that is considered an important indicator of a healthy diet and is associated with a number of health benefits [22,69]. In particular, it is breakfast that ensures the consumption of the most important food products (fruits, vegetables, dairy products), vitamins and minerals and, accordingly, a reduction in the consumption of fats, simple carbohydrates and salt during other meals [4,34,70]. Despite the significant contribution of breakfast to health, most studies reported that from 10% to 30% of children and adolescents never consumed breakfast [4], and the share of teenagers who had a regular breakfast ranged from 45% to 95%, depending on the country [71]. In the developmental period of adolescence, the frequency of skipping breakfast is conventionally associated with body dissatisfaction and attempts to lose weight, dieting and other behaviors oriented at significant adults and idols, and at fashion trends imposing images of slim and thin models. Specific dietary preferences and dietary patterns (vegetarian, vegan, gluten-free and lactose-free, etc.) aimed at weight control [72] may also explain the higher prevalence of this phenomenon in girls [73]. Our findings are consistent with many studies in this field, such as the British study of adolescents aged 11–18 years demonstrating a higher frequency of skipping breakfast among girls and older children. Many of the girls included in that study indicated that they were on a diet [34]. Our data are also consistent with large epidemiological studies covering adolescents in Europe, the United States, Australia, New Zealand, Asia and Africa, where similar trends towards an increase in the number of skipped breakfasts from childhood to adolescence [4,30] were observed, and more so among girls [4,34,52].

### 4.2. Gender-Specific and Age-Specific Features of PA

As the results of our study showed, with an increase in age from 10–12 to 12–15 years, and especially up to 16–17 years, the predisposition of children to poor sleep patterns and reduced PA (PAP) sharply increased. As children enter adolescence, parental control and the control of other significant adults weakens, while peer influence becomes more dominant. New interests increase the number of new contacts with the expansion of their boundaries, which leads to a large expenditure of time. The latter most often changes their daily schedule: sleep time decreases, and the PA level declines [74], which is consistent with the conventional patterns of this period in age-based development. This may be partly due to a shift in near-range rhythms during adolescence towards a later sleep–wake cycle [74]. It is during puberty, between the ages of 10 and 20 years old, that the sleep schedule becomes progressively later. A person’s chronotype, that is, the preference for an early or late sleep–wake cycle, shifts from early to late, and sleep duration decreases at this age [75].

### 4.3. Territorial Dietary Features vs. PA

Territorial, climatic and socioeconomic factors make their own adjustments to the adherence to traditional dietary patterns (DP1) and lifestyle (PAP) of school students. Male adolescents in our study living in Murmansk, compared with Moscow, had a more balanced diet (DP1), including the consumption of more meat, dairy products, fish, fresh fruits and vegetables, and hot meals. This finding may reflect the commitment to home meals, often characteristic of smaller cities. Murmansk is an arctic city with harsher living conditions and a less developed social infrastructure and catering industry, compared with Moscow. It can be assumed that these features of the region dictate healthier traditional lifestyle rules, such as a more orderly daily routine, a more balanced and healthy diet, adherence to family values and conventionally accepted leisure activities (such as extracurricular lessons in various knowledge fields, hobbies, fitness and sports).

It should be noted that the climate of the northern territories is characterized by low temperatures and strong winds, which limits the ability of a child to stay outdoors. However, there were Murmansk teenagers, especially girls, who had higher rates of PA and sleep duration (PAP), in contrast to Moscow school students. The longer sleep duration in Murmansk teenagers, in contrast to their Moscow peers, is apparently dictated by the peculiarities of the light regime in the winter and the region of residence. Higher rates for this pattern (PAP) among adolescents in Murmansk may be due, on the one hand, to the insufficiently developed system of leisure and entertainment complexes, inherent in the Moscow region, and the relatively low living standards of the population in the northern regions, as compared with Moscow residents. On the other hand, small towns in Russia are distinguished by more traditional family relations and ways of spending leisure time (school sports activities, clubs, physical education lessons at school). This hypothesis requires a more detailed and further in-depth study.

### 4.4. Socioeconomic Dietary Features vs. Physical Activity

A low socioeconomic status of the family (single-parent family, lower level of mother’s education, lack of a separate room for a teenager) influences lifestyle and, as shown by the results of our study, negatively affects dietary patterns (DP2, DP3). The formation of eating habits and nutrition culture in youths occurs in the family, and the food preferences of adolescents largely depend on the family diet and traditions. We demonstrated that school students from incomplete families were more prone and significantly more likely to have an unbalanced diet (DP3), which could be due to the lower level of material well-being in these families. As revealed in some studies [76], the latter seriously affects the eating habits and diet in adolescents. As shown by our data, a lower level of the mother’s education significantly increased the likelihood of eating unhealthy foods (DP2), and this was happening from an early age and was apparently connected, among other things, with a lack of awareness of a proper nutrition and a healthy diet. Our results are consistent with most studies from countries with different living standards, showing that socioeconomic status and higher parental education may be directly connected with dietary preferences, especially because higher income also implies higher literacy in issues related to nutrition. Higher income also affects the availability of healthy foods and the motivation to adhere to a healthy lifestyle [55,76,77,78,79,80,81].

The relationship of smoking in the family with the eating habits and lifestyle of school students attracts special attention. The results showed that teenagers in smoking families were to a greater extent prone to DP1 and DP2. In Moscow families, the probability of reducing the consumption of healthy foods and increasing the consumption of unhealthy foods was almost twice as much (OR = 1.81 and OR = 2.42, respectively). In Murmansk, school students and teenagers 13–15 years old in smoking families (OR = 2.07 and OR = 1.74, correspondingly) experienced malnutrition (DP3).

The food preferences of adolescents, as noted above, most often matched the features of the family diet. Studies of the dietary pattern in smoking adults indicated a lower intake of vegetables and fruits [82]. Moreover, the negative effect of nicotine on taste buds, which contributes to the weakening of taste perception, leads to a change in the diet of the smoker via increasing the consumption of smoked meats, canned foods, sausages and salt, thereby enhancing the taste of the consumed food. Therefore, smoking parents form unhealthy eating habits in their children, based on their own food preferences [82,83]. The peculiarities of the eating behavior in school students living in smoking families are in many ways similar to the eating habits of smoking adolescents and are characterized by a smaller incidence of vegetables and fruits in their diet and the frequent consumption of fast food and sugary carbonated drinks [84].

The study [85] revealed that in families where the mother smoked, school-age children were much less likely to eat breakfast regularly, which could have been due to the lack of family traditions of sharing meals and the want of morning time in mothers. It is also possible that smoking mothers often rely on their children eating in cafeterias or at other food outlets, but with the lack of proper nutrition principles and low competence in healthy eating habits, adolescents would undoubtedly focus on peers and more vivid taste sensations that fast food products and sugary drinks provide.

### 4.5. Study Advantages and Limitations

The conducted research is characterized by a number of advantages. For instance, our study was carried out in two substantially different regions of Russia on groups of school students with similar gender and age compositions and living conditions. In Russia, there were no prior studies examining the associations of family, social and regional characteristics of living with the dietary preferences, eating habits and lifestyle of school students. Among the available scientific publications for other countries, such studies are also rare and are mainly devoted to certain issues of the problem under study rather than to all of them at once.

However, our study is not without its shortcomings. Its significant limitation is the small sample size, which, of course, should result in cautious extrapolation of the obtained results to the entire Russian population. Moreover, regions from only two climatic and geographical zones were studied, while the territory of Russia is geographically much more diverse.

Another shortcoming is that the sample of schools in the study is not random, hence requiring careful extrapolation of the study results to the entire population. To ensure comparability of data in cities, the choice of schools was carried out in residential areas that were similar in terms of the socioeconomic living conditions of the population. Moreover, when choosing schools, we took into account the possibility of our study approval by the governments of Moscow and Murmansk, as well as by the school administrations.

The cross-sectional design of the study limits the results in terms of their cause-and-effect relationships.

Another disadvantage of the study is as follows: the analysis of the family environment did not take into account household material wealth, which is among the most important factors affecting lifestyle and quality of life, especially of the younger generation. We attempted to collect such data in our study; however, due to a large number of refusals to specify family income, indicators of wealth were not included in the final analysis.

## 5. Conclusions

The results of our study demonstrated that the behavioral and eating habits of school students largely depend on their gender and age characteristics, family features, social factors and the area of residence. The most significant determinants of a balanced diet, healthy nutrition, rational daily routine and lifestyle in school students were their city of residence, the mother’s education, a complete family and family smoking, along with gender and age characteristics. It is important to point out that a number of associations of dietary and PA patterns in adolescent boys were opposite (in terms of their direction) of those in girls. Identifying the predictors of healthy eating habits and PA in children and youths could help in planning preventive programs to maintain and improve adolescent health and quality of life in the future. Adolescent prevention programs should be aimed not only at teaching healthy eating, but also at changing behavior. These programs should also consider the lifestyle of youths and their social environment. The obtained results implied, among other things, the need to develop gender-oriented programs aimed at promoting a healthy lifestyle among teenagers. Preventive programs for the formation of a healthy lifestyle and preservation and improvement of the health of children and youths should be focused not only on individual work with adolescents, but also on preventive counseling of their parents. Special attention should be paid to working with smoking parents, because smoking in families has a negative impact on the formation of healthy lifestyle skills in children rather than their quality of life alone. It is the eating habits in smoking families that form unhealthy dietary patterns in children and contribute to an imbalance in the nature of their nutrition, which in the future may cause the deterioration of the health profile in younger generations. In addition, healthcare organizers, as well as heads of departments of physical culture and sports, need to take into account the geographical and socioeconomic characteristics of the area where adolescents live. There is no doubt that preventive approaches aimed at improving the health of youths and at the formation of stable stereotypes of a healthy lifestyle in them should be comprehensive, i.e., including activities at the individual, school, family and territorial scales.

## Figures and Tables

**Table 1 ijerph-19-05779-t001:** General characteristics of the sample, % (*n*).

Groups		10–12 y/o	13–15 y/o	16–17 y/o	TOTAL by Gender	TOTAL in Cities
Moscow	Girls	29.9 (66)	40.7 (90)	29.4 (65)	100.0 (221)	100.0 (441)
	Boys	32.7 (72)	36.9 (81)	30.4 (67)	100.0 (220)	
Murmansk	Girls	31.4 (50)	40.9 (65)	27.7 (44)	100.0 (159)	100.0 (342)
	Boys	24.6 (45)	51.4 (94)	24.0 (44)	100.0 (183)	

**Table 2 ijerph-19-05779-t002:** Factor loadings of main identified dietary and PA patterns.

Parameters	Identified Patterns			
	DP 1	DP 2	DP 3	PAP
Fruits and vegetables	0.48	-	-	-
Hot meals	0.48	-	-	-
Meat and meat products	0.64	-	-	-
Fish and fish products	0.45	-	-	-
Milk and dairy products	0.57	-	-	-
Fast food	-	−0.78	-	-
Carbonated drinks	-	−0.78	-	-
Smoked meats and canned food	-	0.52	-	-
Meal frequency	-	-	0.81	-
Availability of breakfast	-	-	0.78	-
Sleep duration	-	-	-	−0.64
Morning exercises	-	-	-	0.73
Physical education lessons	-	-	-	0.42
Outdoor walks	-	-	-	0.32
Explained variance, %	0.15	0.16	0.13	0.31

Note: DP—dietary pattern, PAP—physical activity pattern.

**Table 3 ijerph-19-05779-t003:** Two-way associations of the proportion of children in adverse dietary and physical activity patterns.

Groups		3rd Tertile DP1		3rd Tertile DP2		3rd Tertile DP3		3rd Tertile PAP	
		% (*n*)	*p*	% (*n*)	*p*	% (*n*)	*p*	% (*n*)	*p*
Gender	Girls	32.6 (124)	0.85	23.2 (88)	<0.0001	38.9 (148)	0.0055	36.6 (139)	0.12
	Boys	32.0 (129)		44.2 (178)		29.5 (119)		31.3 (126)	
Age	10–12 y/o	30.5 (71)	0.24	33.5 (78)	0.86	27.9 (65)	0.021	11.6 (27)	<0.0001
	13–15 y/o	30.6 (101)		33.3 (110)		39.1 (129)		40.6 (134)	
	16–17 y/o	36.8 (81)		35.4 (78)		33.2 (73)		47.3 (104)	
City	Moscow	33.8 (149)	0.32	33.6 (148)	0.78	32.4 (143)	0.26	36.5 (161)	0.074
	Murmansk	30.4 (104)		34.5 (118)		36.6 (124)		30.4 (104)	
Room	Separate	30.6 (184)	0.065	34.3 (206)	0.74	32.8 (197)	0.16	34.3 (206)	0.64
	Shared	37.9 (69)		33.0 (60)		38.5 (70)		32.4 (59)	
Family	Incomplete	32.1 (52)	0.95	40.1 (65)	0.063	42.0 (68)	0.018	40.1 (65)	0.058
	Complete	32.4 (201)		32.4 (201)		32.0 (199)		32.2 (200)	
Number of children	1 child	31.7 (120)	0.71	35.6 (135)	0.35	31.7 (120)	0.16	32.7 (124)	0.52
	2 or more	32.9 (133)		32.4 (131)		36.4 (147)		34.9 (141)	
Mother’s education	Not higher	34.1 (74)	0.51	37.8 (82)	0.16	37.8 (82)	0.18	29.9 (65)	0.15
	Higher	31.6 (179)		32.5 (184)		32.7 (185)		35.4 (200)	
Smoking	No	28.9 (141)	0.0099	29.8 (145)	0.0015	31.6 (154)	0.061	34.7 (169)	0.52
	Yes	37.8 (112)		40.9 (121)		38.2 (113)		32.4 (96)	

**Table 4 ijerph-19-05779-t004:** Effect of gender, age, area of residence and social conditions on the likelihood of adverse patterns.

Groups		3rd Tertile DP1		3rd Tertile DP2		3rd Tertile DP3		3rd Tertile PAP	
		OR	95% CI	OR	95% CI	OR	95% CI	OR	95% CI
Gender (reference: girls)	Boys	1.00	0.74–1.36	2.72 *	1.97–3.75	0.65 *	0.48–0.88	0.78	0.56–1.07
Age (reference: 10–12 y/o)	13–15 y/o	1.06	0.72–1.55	0.74	0.71–1.63	1.63 *	1.12–2.38	5.69 *	3.55–9.13
	16–17 y/o	1.36	0.90–2.00	0.99	0.68–1.45	1.20	0.79–1.82	7.34 *	4.45–12.10
City (reference: Moscow)	Murmansk	0.89	0.65–1.22	1.00	0.73–1.38	1.16	0.85–1.59	0.71 *	0.51–0.99
Room (reference: separate)	Shared	1.40	0.98–1.99	1.01	0.70–1.48	1.23	0.86–1.76	0.90	0.61–1.32
Family (reference: incomplete)	Complete	1.08	0.74–1.60	0.70	0.47–1.02	0.66 *	0.45–0.96	0.71	0.48–1.05
Number of children (reference: 1)	2 or more	0.92	0.67–1.27	0.77	0.56–1.07	1.22	0.89–1.68	1.01	0.73–1.40
Mother’s education (reference: not higher)	Higher	0.87	0.62–1.24	0.69	0.48–0.98	0.85	0.61–1.20	1.02	0.71–1.48
Smoking (reference: none)	Yes	1.47 *	1.07–2.03	1.64 *	1.18–2.27	1.27	0.93–1.75	0.74	0.53–1.03

Note: In all models, adjustments were made for individual indicators: gender, age, city, availability of a separate room, family composition, number of children in the family, mother’s education, parents’ smoking; * *p* ≤ 0.05.

## Data Availability

The data presented in this study are available on request from the corresponding author. The data is not publicly available due to the need to maintain the confidentiality of study participants.

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
