# Peer review of "Gender, Age, Family and Territorial Features of Dietary and Physical Activity Patterns in Russian Youths"

_ijerph, 2022, doi:10.3390/ijerph19095779_

Round 1
Reviewer 1 Report
The paper Gender, age, family and territorial features of dietary and
physical activity patterns in Russian youthsdeals with a real and important topic, which is consistent with the purpose of the journal.
However, it needs revisions before it can be considered for publication in the journal. The following comments should be considered.
1. It would be necessary to have a literature review in which to present the concepts studied and the results of previous studies regarding the influence of gender, age, family and territorial features of dietary and physical activity on people, the link between the variables should be presented much more clearly.
2. The research methodology should be presented more clearly, the period when the research took place, the sampling method used, the objectives and hypotheses of the research, the variables independent and dependent
3. In the discussion section the results obtained could be compared with the results of other research carried out, possibly in other countries, and the opinion of the authors on similarities or differences should be presented.
4. The conclusions section should be developed by introducing theoretical managerial implications
Reviewer 2 Report
This study carried the research on Identifying dietary and physical activity (PА) patterns in Russian youths and 8 examining their dependence on gender, age, family characteristics and area of residence features. With the survey in Moscow and Murmansk, the authors carried out the empirical research. Generally speaking, this study is interesting and meaningful. However, there are several comments as follows.
- The data presented in this paper is not clear enough, one list table is necessary to introduce the dietary measures and other important variables.
- It is also necessary to present all the results of the principal analysis and related information that would be crucial for the results assessment.
- One of my big concerns is the model adopted in the paper is also not clear enough. In 2.4 Statistical data processing, the authors just simply indicated that the logistic regression is used. The problem is that how to define the favorable patterns is unclear. To my mind, the author employed the principal analysis and finally got the principal factors which are, obvious, non-discrete variables.
- Another issue related the principal factors are the direction of each variable. Some of them are positive and others are not, when combining they together, is it safe to presume the larger the principal the better or on the contrary
- There may be some cross relations among the dietary pattern and physical activity pattern which would also need to take account in the model and discussions.
Reviewer 3 Report
It is a meaningful paper that analyzes dietary and physical activity patterns. For revision, it is necessary to consider the following.
First, the independent variables affecting the two dependent variables are not based on the theoretical background. Theoretical discussions on each set independent variable should be organized. In particular, in addition to demographic variables, it is necessary to establish independent variables that can be of interest from a theoretical level.
Second, the practical implications of the research results should be presented. In the conclusion section, not only a summary of the study, but also the implications of this study should be extensively added.
Third, the sample does not follow probability sampling. These limitations need to be discussed.
Reviewer 4 Report
Dear authors, thank you for the invitation to review the article, the article is very interesting and very well organized.
I just have a few concerns:
Considering the detailed information with which the different diet patterns were described, the same information on physical activity would be beneficial. How many times per day, per week, what the sports and other important information if you have this info, please add.
Round 2
Reviewer 1 Report
Dear authors, the efforts to improve the article are obvious. The conclusions could be filled up with more recommendations. I recommend publishing the article after minor revisions
Reviewer 2 Report
Most of the concerns were respnosed. and the revision was improved. Since the observations were grouped into tertiles depending on the individual commitment of the children, it is better to simply summarize the differences between the groups.
Reviewer 3 Report
Well revised
